# The Role of Glp-1 Receptor Agonists in Insulin Resistance with Concomitant Obesity Treatment in Polycystic Ovary Syndrome

**DOI:** 10.3390/ijms23084334

**Published:** 2022-04-14

**Authors:** Krzysztof Bednarz, Karolina Kowalczyk, Marlena Cwynar, Dominika Czapla, Wiktor Czarkowski, Dominika Kmita, Artur Nowak, Paweł Madej

**Affiliations:** 1Department of Gynecological Endocrinology, Medical University of Silesia in Katowice, 40-752 Katowice, Poland; kbednarz9718@gmail.com (K.B.); marlenac.87@gmail.com (M.C.); dominika.czapla@op.pl (D.C.); czarkowski.wiktor@gmail.com (W.C.); domminika.kmita@gmail.com (D.K.); pmadej@sum.edu.pl (P.M.); 2Gynecological and Obstetrician Polyclinic, District Hospital, 15-435 Białystok, Poland; artur_nowak@mp.pl

**Keywords:** polycystic ovary syndrome (PCOS), insulin resistance (IR), glucagon-like peptide-1 receptor agonists (GLP-1RAs), obesity, inflammation, oxidative stress, lipid metabolism, infertility

## Abstract

Insulin resistance is documented in clamp studies in 75% of women with polycystic ovary syndrome (PCOS). Although it is not included in the diagnostic criteria of PCOS, there is a crucial role of this metabolic impairment, which along with hormonal abnormalities, increase each other in a vicious circle of PCOS pathogenesis. Insulin resistance in this group of patients results from defects at the molecular level, including impaired insulin receptor-related signaling pathways enhanced by obesity and its features: Excess visceral fat, chronic inflammation, and reactive oxygen species. While lifestyle intervention has a first-line role in the prevention and management of excess weight in PCOS, the role of anti-obesity pharmacological agents in achieving and maintaining weight loss is being increasingly recognized. Glucagon-like peptide-1 receptor agonists (GLP1-RAs) not only act by reducing body weight but also can affect the mechanisms involved in insulin resistance, like an increasing expression of glucose transporters in insulin-dependent tissues, decreasing inflammation, reducing oxidative stress, and modulating lipid metabolism. They also tend to improve fertility either by increasing LH surge in hypothalamus-pituitary inhibition due to estrogen excess connected with obesity or decreasing too high LH levels accompanying hyperinsulinemia. GLP1-RAs seem promising for effective treatment of obese PCOS patients, acting on one of the primary causes of PCOS at the molecular level.

## 1. Introduction

Polycystic ovary syndrome (PCOS) is a significant health burden that impairs women’s quality of life [1]. It is a common condition [1], but its precise worldwide presence is very difficult to evaluate due to the lack of a universal definition for this disease, diversity of its phenotypes, the possible influence of age, ethnicity [2], and different diagnostic criteria usage [3,4]. For these reasons, the prevalence of PCOS ranges from 2.2% to 26% in different studies [5]. Although the exact reason for PCOS is not yet discovered, there are multiple genetic [3,6] and environmental factors that play an important role in the occurrence of this disorder [7,8].

The essence of PCOS is a heterogenous presentation of androgen excess (which appears as hyperandrogenism, hirsutism, acne, alopecia, and seborrhoea), ovulatory dysfunction (which appears as oligo-ovulation, menstrual dysfunction, and subfertility) and polycystic ovarian morphology (PCOM). It is believed that hyperinsulinemia is the main reason for increased androgen secretion because insulin acts as a co-gonadotropin on the ovary, facilitates androgen secretion from the adrenal glands, and modulates releasing of luteinizing hormone (LH) [8]. Because of these reasons, women with increased insulin levels (regardless of their endo-or exogenous origin) have increased prevalence of PCOS. Androgen excess by favoring visceral adiposity contributes to insulin resistance (IR) and to the development of hyperinsulinism [9]. As studies show, visceral adipose tissue in women with PCOS differs from that in healthy women, and in genomic, transcriptomic, and proteomic profiles is like male visceral adipose tissue. However, the fact that PCOS does not appear in every woman with IR or hyperinsulinemia and the fact that not every woman suffering from PCOS presents IR, suggests that there must be another malfunction that favors androgen excess in response to insulin or other triggering factors [8].

Depending on the presenting features, there are three different clinical manifestations of PCOS. The classic phenotype consists of hyperandrogenism and oligo-ovulation. It is linked with the most severe insulin resistance and metabolic comorbidities. The next phenotype is ovulatory PCOS which comprises hyperandrogenism and PCOM. It co-occurs with moderate IR and metabolic dysfunction. The least severe is the non-hyperandrogenic phenotype, which presents with oligo-ovulation, PCOM and has the weakest connection with IR and metabolic comorbidities [8].

Treatment of PCOS should be adjusted to the individual needs of the patient’s complaints and symptoms [10]. Androgen excess, IR, obesity, and oligoovulation might be the targets of treatment [10]. Early prevention of metabolic consequences needs to be highlighted. Hyperglycemia, obesity, hypertension, and dyslipidemia define the metabolic syndrome, which occurs with PCOS more often [11]. Therefore, women with PCOS must be regularly screened for obesity (using BMI and waist circumference), blood pressure, glucose, and lipid profile [12]. The excess nutrients cause hypertrophy or hyperplasia of adipocyte and induces IR, hyperinsulinemia [11], higher serum free fatty acids, lipogenesis, and increased fat storage in the liver, pancreas, and skeletal muscles [13]. In consequence, glucose tolerance disorders lead to type-2 diabetes mellitus (T2DM) [12].

Education, counseling, and a healthy lifestyle, including appropriate diets and physical activity, should be the focus of the therapeutic proceedings [1,14]. A low-glycemic load (20 to 40 g/day), as well as low-fat diets (40 g of fat per day), were confirmed to be effective for weight control but not for biochemical hyperandrogenism. There is also no indication that a low-glycemic loan diet was better than a low-fat diet for attenuating hyperandrogenism [15]. There is not enough strong evidence that any specific diet leads to better clinical outcomes. Diet should be managed individually and should include energy deficit to reduce weight. Regular healthy eating and physical exercise improve general health conditions and optimize hormonal balance [1]. Losing 5–10% of body weight is considered a significant clinical improvement, especially among patients with excess weight [1]. A minimum of 250 min/week of moderate intensity or 150 min/week of vigorous intensity activities or an equivalent combination of both, and muscle-strengthening activities involving major muscle groups on two non-consecutive days/week are recommended [1]. However, among obese women, joints problem and arthritis could be significant limitations to exercise. Therefore, physical activity must be tailored to personal abilities [16]. Lifestyle interventions should be supported by interactions with a mental health professional and should include psychological care to improve a patient’s emotional state. It is worth noting that anxiety and depression relatively often coexist with PCOS. The research confirmed depressive symptoms in 44% and anxiety disorder in 41.9% of women with PCOS [1]. Lower mood, loss of motivation co-occurring with negative body image usually causes the lifestyle change to be a real challenge for patients. Women with PCOS are more likely to follow weight management practices [17], although they also consider achieving and maintaining this reduction challenging [18]. Moreover, there is evidence that suggests that women with PCOS may encounter more barriers and difficulties in losing weight due to reduced utilization of lipid stores [19] and decreased brown adipose tissue activity [20]. There is also a lack of data describing long-term compliance among patients with PCOS with the dietary regime and lifestyle modifications, which translates into the effectiveness of the therapy. Although even in well-supported programs such as lifestyle intervention in randomized controlled trials, there was a high risk of dropping out during long-term follow-up [21], and investigators report high attrition rates [22,23]. It is considered that among adolescents with PCOS, only around 60% adhere to nutritional interventions [24].

On the other hand, more radical methods of treatment, like bariatric surgery and specific medicaments, can be considered. According to guidelines for obesity management [25], bariatric surgery is recommended for patients with BMI ≥ 35 kg/m^2^ and a minimum of one severe complication. Evidence in the treatment of PCOS is limited, and there is no recommendation available. This experimental therapy can induce malabsorption and eating disorders. It could also impact future pregnancies and the fetus, and it increases neonatal mortality [26]. Anti-obesity medications can be a consideration for adults with PCOS. However, due to the possible side effects, as well as variable availability, it is recommended to avoid pregnancy as long as such medications are being taken [1]. Reduction of obesity, as well as testosterone levels, and as a result, improving ovulation, can be achieved by using metformin. It decreases gluconeogenesis, lipogenesis, and enhances uptake of glucose in the skeletal muscle, liver, and adipose tissue. It should be recommended in PCOS with BMI ≥ 25 kg/m^2^ in addition to a healthy lifestyle [27], particularly for patients with disturbed glucose tolerance [28]. Different diabetic medicines like the insulin-sensitizing treatment in PCOS-glucagon-like peptide-1 receptor agonists (GLP-1RAs), dipeptidyl peptidase 4 (DPP-4) inhibitors, and sodium-glucose cotransporter (SGLT2) inhibitors could also be considered as a part of the treatment of PCOS [29,30].

GLP-1RAs release glucose-dependent insulin from the pancreatic cells. The main action of GLP-1RAs is summarized in the diagram (Figure 1). Using GLP-1RAs leads to a reduction in weight, testosterone levels, and improves ovulation among obese patients [31]. Our review is focused on the role of the GLP-1RAs in the treatment of PCOS. The paper presents current data on the possible use of GLP-1 agonists in PCOS therapy, based on articles from 2010 to 2022.

## 2. The Pathomechanism and Role of IR in PCOS

IR is an impairment of insulin effect on cells, especially muscle cells, adipocytes, and hepatocytes. It leads to carbohydrate, lipid, and protein metabolism alterations. IR is an important element of metabolic disorders, including T2DM and pre-diabetes (impaired glucose tolerance and impaired fasting glucose) [32]. In PCOS, IR is present in obese and overweight patients, as well as in those of normal weight, although in the latter group, with a lower frequency (59.3% in normal weight vs. 77.5% in overweight and 93.9% in obese patients) [33]. This difference is probably the result of an increase in adipose tissue volume, which plays a crucial role in the pathomechanism of IR. The incidence of IR is also different according to the phenotype of PCOS. It has been shown that IR is present more often in the classic phenotype (80.4%) than in the ovulatory (65.0%) and normoandrogenic (38.1%) [34]. Moreover, it is more frequent in patients with anovulatory compared to those with ovulatory cycles in PCOS [35].

The mechanism of IR in PCOS is complex and still not fully understood. Increased adipose tissue volume, especially visceral adipose tissue, plays an important role in IR development [9]. It is influenced by many factors, including excessive energy intake and high androgen levels that promote a change in body fat distribution with an increase in visceral fat volume [9,36]. In PCOS, hypertrophic adipocytes are associated with metabolic disorders [37]. In a study on adipocytes from healthy women, it was observed that testosterone, acting through its receptor, induces IR in these cells by affecting insulin-stimulated phosphorylation of protein kinase C (PKC) (Figure 2) [38].

In over-expanded adipose tissue, the profile of adipokines secreted by adipocytes is impaired. As adipose tissue volume increases, the concentration of adiponectin, having an insulin-sensitizing and anti-inflammatory impact, declines. Its lower concentration is associated with the development of IR and T2DM [39]. Adiponectin reduces lipid storage in the liver and muscles, which results in decreased plasma membrane sn-1,2-diacylglycerol-induced PKC activity and increases insulin signaling. Adiponectin mediates these effects by promoting the storage of triglyceride (TG) in white adipose tissue through stimulation of lipoprotein lipase and by increasing muscle fat oxidation and glucose uptake by stimulation of 5′ adenosine monophosphate-activated protein kinase (AMPK) in muscle (Figure 3) [40]. Moreover, in PCOS, adiponectin levels are decreased in both obese and nonobese patients [41,42]. Changes in other adipokines, such as apelin, vaspin, resistin, and chemerin, are also observed in women with PCOS. Apelin is adipokine expressed in adipocytes, and ovarian cells, and its expression is stimulated by insulin. It increases the secretion of progesterone and estradiol in granulosa cells in response to stimulation by insulin-like growth factor-1 (IGF-1). Vaspin, an adipokine expressed mainly in visceral adipose tissue and in the ovary, enhances granulosa cells’ steroidogenesis and proliferation. The expression of resistin is upregulated by androgens, therefore there is a positive association between hyperandrogenism and high levels of resistin in PCOS patients. Resistin may be involved in the pathogenesis of IR in PCOS. Chemerin is expressed in ovarian cells and plays an important role in adipose tissue inflammation [43].

Adipose tissue growth without adequate vascularization results in hypoxia. Reduced adipose tissue oxygenation contributes to IR by decreasing branched-chain amino acid catabolism and increasing plasma branched-chain amino acid concentrations [44]. Chronic inflammation also develops with increased levels of tumor necrosis factor α (TNF-α), monocyte chemotactic protein-1 (MCP-1), C-reactive protein (CRP), interleukin-1 (IL-1), interleukin-6 (IL-6), and other pro-inflammatory cytokines, which have been found elevated in PCOS. Additionally, hyperandrogenism can increase inflammation through activation of the nuclear factor kappa B (NF-κB) inflammation pathway [45,46]. Hormones, adipokines, and cytokines associated with inflammation affect the intracellular enzymatic pathways associated with the insulin receptor. Insulin acts through binding and activation of insulin receptors type A and B (IR-A and IR-B) located in the cell membrane. Activation of receptor subunits, which are glycoproteins with tyrosine kinase activity, leads to phosphorylation of the insulin receptor substrates (IRS-1 and IRS-2). Then, the signal is transmitted through intracellular signalling pathways. The phosphoinositide 3-kinase/protein kinase B (PI3K/Akt) pathway is involved in carbohydrate metabolism by increasing intracellular glucose transport through glucose transporter type 4 (GLUT-4) in muscle cells and adipocytes, inhibiting hepatic gluconeogenesis and glycogenolysis, and promoting glycogenesis. The mitogen-activated protein kinase (MAPK) pathway is involved in promoting proliferation and cell growth [47]. Defects in insulin signal transduction play a crucial role in the pathomechanism of IR. It is a result of protein modifications, such as increased IRS-1 serine phosphorylation, which affects metabolic pathways in many tissues, including the ovary [48]. TNF-α affects insulin signaling by phosphorylation of serine in IRS-1 through activation of several serine kinases, including c-Jun-NH2-terminal kinase (JNK) and extracellular signal-regulated kinase (ERK), which inhibits insulin-induced tyrosine phosphorylation of IRS-1 and downregulates PI3K activity [49]. It causes decreased GLUT-4 expression and decreased glucose transport into cells (Figure 2). Moreover, mitochondrial damage, increased oxidative stress, and endoplasmic reticulum stress are also involved in the development of IR [50]. Metabolic inflexibility is the impaired ability to switch from lipid to carbohydrate oxidation under insulin-stimulated conditions. This state is associated with IR. However, some data show that PCOS women have normal metabolic flexibility, which could suggest a distinct pathophysiological mechanism for IR in this group [51].

Studies on mice have shown that activation of androgen receptors by dihydrotestosterone in the brains of the tested animals led to the development of IR, whereas rodents deprived of these receptors did not develop this type of disorder. That observation suggests that androgens may also promote IR by acting on the central nervous system [52]. Moreover, prenatal exposure to high levels of androgens can probably lead to the development of IR and PCOS [53,54,55].

The role of other factors in pathomechanism of IR is also under investigation. MicroRNA (miRNA) are about 22 nucleotides long non-coding RNA molecules that are responsible for post-transcriptional regulation of gene expression [56]. Studies show that these particles, located in exosomes of adipocytes and macrophages of adipose tissue and secreted from them into the circulation, can influence the development of IR through mechanisms, such as inhibition of peroxisome proliferator-activated receptor (PPAR-gamma) expression and affecting PI3K/AKT-GLUT4 signaling pathway [57,58,59]. An altered profile of miRNA particles in plasma has been observed in PCOS [60]. As a result, miRNAs can be used as diagnostic markers and, in the future, be important in the therapy of PCOS and IR.

The result of all the processes described above is impaired glucose transport into cells by insulin-dependent membrane GLUT 4 transporters, reduced glycogen synthesis and decreased inhibition of hepatic gluconeogenesis. This leads to hyperglycemia and to compensatory hyperinsulinemia. IR, the resulting hyperinsulinemia, and increased androgen levels influence each other, and it is difficult to determine which phenomenon occurs first (Figure 4) [61]. Insulin acting on ovarian theca cells increases androgen synthesis by increasing activity of the cytochrome P450C17α, which has 17-hydroxylase and 17,20-lyase activities [62]. Insulin acts on ovarian theca cells by further increasing LH stimulatory effect [63]. Moreover, insulin affects the hypothalamic-pituitary system. Through MAPK and increased gonadotropin-releasing hormone (GnRH) gene expression, it increases the frequency and amplitude of GnRH release pulses and, in consequence, LH release, which stimulates ovarian cells to androgen synthesis [64]. Insulin also upregulates adrenal androgen synthesis by affecting steroidogenic factor 1 (SF-1), which is a transcriptional factor playing an important role in steroid hormone synthesis by regulating the transcription of steroidogenic genes, including StAR, Cyp11a1, Cyp17, Cyp11b1, Cyp11b2, and 3β-Hsd [65]. Moreover, insulin decreases the release of sex hormone-binding globulin (SHBG) from the liver, thus increasing the amount of the free, non-protein-bound androgen fraction, which exerts biological effects on tissues. Studies indicate that SHBG can be useful as a marker for a higher risk of PCOS [66].

## 3. Role of GLP-1RAs in IR Associated with PCOS

The GLP-1RAs, also known as incretin mimetics, are a group of drugs that imitate the activity of the incretin hormone GLP-1 secreted in the distal part of the small intestine [67]. This hormone binds to insulin receptors on the beta cells and stimulates insulin secretion [67,68,69]. Moreover, this secretion is stimulated by GLP-1 only, then glucose levels increase [70]. The GLP-1 does not demonstrate activity when blood glucose levels are normal or in cases of hypoglycaemia [70]. The incretin mimetics stimulate insulin secretion, decrease glucagon secretion, inhibit the hunger center, and delay gastric emptying, thereby the postprandial glucose levels improve, and the feeling of satiety appears earlier [69] (Figure 1). They are resistant to the DPP-4 inhibitors. The GLP-1RAs decrease blood glucose levels, but they do not cause hypoglycemia [68]. Furthermore, GLP-1RAs are metabolically beneficial because they decrease low-density lipoprotein (LDL) and TG levels [69]. The GLP-1RAs treatment reduces body weight and diminishes the risk of cardiovascular disease (CVD) [71,72].

Currently, due to beneficial effects, incretin mimetics are applied in T2DM and treatment of obesity. The GLP-1RAs therapy enables a significant reduction in body weight [72,73]. According to American Diabetes Association 2022 recommendations, GLP-1RAs with or without metformin are appropriate initial treatment for patients with T2DM with or at high risk for atherosclerotic cardiovascular disease, heart failure, and/or chronic kidney disease [74]. Incretin mimetics therapy has a profitable influence on non-alcoholic fatty liver disease (NAFLD) and neurodegenerative diseases like Alzheimer’s disease or Parkinson’s disease [75,76,77,78,79]. The GLP-1RAs also have positive effects in patients with PCOS decreasing IR and improving metabolic alterations [80,81,82,83].

### 3.1. Body Weight Reduction

Obesity, a global pandemic, is a key component of developing hyperinsulinemia, which plays a fundamental role in the pathomechanism of PCOS [84]. Higher circulating levels of insulin increase ovarian androgen production, which has a strong influence on PCOS symptoms [84,85]. Furthermore, adipocytes are responsible for the functional alteration of the hypothalamic-pituitary-ovarian (HPO) axis due to aromatization of androgens to estrogens, which by negative feedback loop, suppress gonadotropin production [84,86]. Additionally, exacerbation of the symptoms of PCOS is associated with the concomitant presence of obesity, which further increases hyperinsulinemia [84,87]. Based on several clinical studies documenting the effects of liraglutide, a long-lasting GLP-1RAs, on weight loss, it has been approved for the treatment of obesity at a dose of 3.0 mg/day in the form of subcutaneous injections [88]. It might exert a significant impact on insulin levels and PCOS symptoms.

Both peripheral and central mechanisms were found to be associated with the weight-loss effects of liraglutide [88]. Knudsen et al. examined the human and non-human studies used to determine the cellular localization of GLP-1 receptor (GLP-1R) and investigate pharmacological effects of GLP-1RAs. GLP-1R is expressed in the pancreas, gastrointestinal tract, heart, lungs, kidneys, and brain, and the localization of these receptors determines GLP-1RAs effects [89]. Drucker identified GLP-1R in islet and pancreatic exocrine cells, the autonomic and enteric nervous systems, blood vessels, Brunner’s glands, and sinoatrial node [90]. GLP-1R populations in the gastrointestinal tract and brain are perceived as valid in the weight-loss effect of GLP-1RAs.

Long-term negative energy balance achieved by a reduction in appetite and energy intake, increase in energy expenditure, or both are required to achieve body weight loss [72]. It has been known that GLP-1 affect weight loss principally via reducing energy intake [89]. In addition, GLP-1RAs promote fullness and satiety [72]. Van Bloemendaal et al. have observed that weight loss facilitated by GLP-1RAs is associated with a reduction in total body fat, particularly trunk or visceral fat [72].

GLP-1RAs exert the effect of regulating gastrointestinal (GI) motility via GLP-1R in the myenteric plexus neurons [89]. In addition, GLP-1R were identified in afferent sensory vagal neurons in the proximal gastrointestinal tract-stomach and duodenum. Considering GLP-1R expression on the vagal nerve, it has been proposed that GLP-1RAs affect weight loss via gut-to-brain communication. However, in vagotomized animals, the effect of liraglutide on body weight has not been compromised, supporting the hypothesis that gut-to-brain communication via the vagal nerve is not the main mechanism responsible for the weight loss [89]. Activation of GLP-1R on vagal neurons increases gastric pressure [90]. Hence, gastric emptying has been proposed as a mechanism of weight loss by GLP-1RAs [89]. However, there are vagal GLP-1R sensory neurons that do not exhibit acute responses to liraglutide and semaglutide, therefore this seems unlikely to be the main mechanism by which GLP-1RA affects weight loss [90].

GI motility promotes the most common adverse effects of GLP-1RAs-transient nausea and vomiting [89]. Respectively, a hypothesis of GLP-1RAs side effects to be involved in weight-loss effect has been explored. A large clinical trial data set with GLP-1RAs has shown that many patients have weight loss and no side effects, hence, GI side effects are not the main mechanism of weight loss [89]. Those findings suggest a further mechanism of action in GLP-1RAs-related body weight loss.

Studies support the concept that GLP-1R in the brain is mainly responsible for decreased food intake and further positive effect of GLP-1RAs on weight loss [89].

Feeding behavior is influenced by homeostatic and non-homeostatic factors. Homeostatic feeding promotes stability in the amount of energy stores and it controls energy balance by adjusting food intake. It is regulated by hunger and satiety. Important CNS structures involved in homeostatic feeding are the brainstem and hypothalamus. The brainstem area postrema (AP) and nucleus tractus solitarii (NTS) transmit peripheral signals (changes in nutrients, hormones, and neuropeptides) to the arcuate nucleus (ARC) of the hypothalamus. Homeostatic feeding is collectively regulated by orexigenic and anorexigenic neurons. Orexigenic neurons express neuropeptide Y (NPY) and agouti-related peptide (AgRP), whereas anorexigenic neurons express pro-opiomelanocortin (POMC). Non-homeostatic feeding behavior is regulated by reward, cognitive, emotional factors, and motivation related to foods. Hence, corticolimbic circuits (including the striatum, amygdala, insula, nucleus accumbens (NAc), and orbito-frontal cortex) affect non-homeostatic eating [72]. In addition, GLP-1R expression was localized in the ventral tegmental area (VTA), and nuclei embedded within the mesolimbic reward circuitry, next to NAc [91]. Both homeostatic and non-homeostatic factors modulate feeding behavior and pathways of these modulators comprise multiple interconnected brain regions [72].

Liraglutide enters the brain in a GLP-1R-dependent manner, and a fluorescent version of liraglutide was observed in the hypothalamus (ARC, paraventricular nucleus (PVN), supraoptic NTS, supraoptic decussation) and several circumventricular organs (CVOs, e.g., median eminence (ME), area postrema (AP), subfornical organ, and organum vasculosum of lamina terminalis). GLP-1RAs do not broadly cross the blood-brain barrier, but they can access neurons in the CVO’s and in a few select sites in the hypothalamus [89].

GLP-1RAs administration reduced feelings of hunger, and it was associated with an increase in functional connectivity of the NTS with the hypothalamus. GLP-1RAs bind to GLP-1R on POMC and cocaine-and amphetamine-regulated transcript (POMC/CART) of ARC neurons. Subsequently, GLP-1 directly stimulates POMC/CART neurons and indirectly inhibits NPY and AgRP and increases measures of satiety, and decreases hunger [92]. These effects via homeostatic feeding regulations lead to reduced energy intake (Figure 5).

Activation of GLP-1R on VTA and NAc decreases reward-motivated behaviors for palatable food. Intra-VTA and intra-NAc injections of GLP-1RA (exendin-4) reduced intake of regular chow, high-fat diet, and body weight. Activation of central GLP-1R was shown to increase tyrosine hydroxylase levels, which is responsible for dopamine synthesis in the VTA associated with inhibition of NAc and cortex-projecting dopamine neurons [91].

Mechanism of action of GLP-1RAs in the CNS provides evidence that GLP-1RAs treatment is associated with a reduction in appetite, hunger, and energy intake. Thanks to alterations in the food reward pathway, they are supposed to decrease food cravings, preference for energy-dense foods and also prevent overeating [91].

Many clinical trials check whether the effect at the cellular level translates into an improvement in the clinical condition of patients.

The research on the impact of liraglutide on ectopic fat in the group of 72 women with PCOS, with a BMI > 25 kg/m^2^ and/or IR showed that 26 weeks of liraglutide treatment decreased body weight by 5.2 kg and pointedly reduced liver fat content, visceral adipose tissue, and the prevalence of NAFLD. Furthermore, liraglutide increased SHBG levels by 19% and decreased free testosterone by 19%. HbA1c, fasting glucose, and leptin also decreased, but there was no influence on measures of IR, glucagon, and adiponectin [93]. The incidence of periods improved (bleeding ratio 0.28 vs. 0.14 placebo) and ovarian volume was reduced by 1.6 mL [82].

A trial with 30 women that compared the impact of a high dose of liraglutide to metformin combined with low dose of liraglutide on weight loss showed that after 12 weeks, both treatment schedules brought considerable weight loss (−6.3 ± 3.7 kg monotherapy vs. −3.6 ± 2.5 kg combined therapy) and improved outcome of OGTT. Liraglutide treatment had a better influence on BMI and waist circumference reduction than combined therapy, but the latter significantly decreased total testosterone and caused less nausea [94].

In a trial with 32 obese women with PCOS treated with liraglutide or metformin for 12 weeks, the weight loss and decrease of BMI were comparable in both groups, but short-term treatment with liraglutide caused superior weight loss than with metformin in the group of patients who had newly diagnosed PCOS, and their risk of the metabolic syndrome was higher (mean BMI reduced by 2.13 kg/m^2^ vs. 0.62 kg/m^2^) [95].

The study concerning the impact of liraglutide on eating behavior was performed on a group of 36 women pre-treated with metformin. After 12 weeks, it turned out that liraglutide decreased the uncontrolled eating score and emotional eating score, which additionally influenced body weight loss (3.8 ± 0.1 kg) [96].

The group of 57 obese women with PCOS took part in research investigating the role of genetic variability in body weight loss. The study showed that the transmitters of a minimum of one polymorphic rs10305420 allele have lower weight loss during liraglutide therapy [97].

The research comparing the efficacy of 12 weeks of liraglutide or roflumilast (inhibitor of phosphodiesterase 4) therapy over a metformin treatment was performed in 45 obese women with PCOS. Liraglutide reduced body weight the most (3.1 ± 3.5 kg), furthermore decreased the visceral adipose tissue area and improved OGTT outcomes [98].

### 3.2. Anti-Inflammatory Properties of GLP-1RAs

One of the crucial factors involved in the pathogenesis of IR in PCOS is chronic inflammation. It is a result of an excessive amount of adipose tissue, especially visceral adipose tissue, and the pro-inflammatory effect of high androgen levels [99]. Therefore, in the therapy aimed at reducing IR in PCOS, drugs with the ability to reduce inflammation should be used. Studies have shown that GLP-1 and its analogs have anti-inflammatory properties.

One of the first stages of inflammation in adipose tissue is the migration of monocytes from blood vessels to the extravascular space as a result of MCP-1 secretion by adipocytes. Monocytes transform into macrophages, which are involved in the inflammatory reaction. This migration is especially intensive in the case of obesity [100]. A mouse model study showed that GLP-1 can decrease macrophage activity in adipose tissue-the expression and production of IL-6, TNF-α, MCP-1, and NF-κB together with JNK activation [101]. In a cell line study, exendin-4, which is a GLP-1RAs, has shown the ability to inhibit the LPS-induced migration of monocytes. Exendin-4 also inhibited the secretion of pro-inflammatory cytokines, including TNF-β, IL-6, and IL-1β, in macrophages by inhibiting the activity of NF-κB. It contributed to increased insulin-stimulated glucose uptake [102].

It was also observed that exendin-4 can increase the production and secretion of adiponectin in a culture medium of 3T3-L1 adipocytes and in high-fat diet-fed mice, which was seen as an increased level of adiponectin mRNA [103]. The protein kinase A (PKA) and Sirt1/Foxo-1 signaling pathways are involved in the effect of exendin-4 on adiponectin secretion in adipocytes [103,104]. Adiponectin has anti-inflammatory and insulin-sensitizing properties [40].

The anti-inflammatory effect of GLP-1RAs is not limited to adipose tissue. PCOS and IR are risk factors for damage to other tissues, including the vascular endothelium [105]. In a study on cultured human aortic endothelial cells (HAECs), stimulated by TNF-α and LPS, liraglutide reduced monocyte adhesion to endothelial cells by reducing expression of VCAM-1 and E-selectin. Liraglutide activated calmodulin dependent protein kinase-b (CaMKKb) by increasing intracellular calcium levels. CaMKKb phosphorylated AMPK and CaMK1 activated next enzymes, including endothelial nitric oxide synthase (eNOS) and cAMP response element-binding protein (CREB), which contributed to anti-inflammatory effect [106]. The study on the effect of liraglutide treatment on atherothrombotic risk carried out on 19 obese women with PCOS and in a control group (17 people) showed that after six months liraglutide caused weight loss (3%, 4%) and decreased atherothrombosis markers like: Inflammation, endothelial function, and clotting [83].

Liraglutide also exerts anti-inflammatory effect on the liver. Studies confirmed that PCOS, especially with high serum total testosterone and free androgen index, presents a higher risk of NAFLD, which is also associated with obesity and metabolic disorders, including IR [107,108]. A study on mice treated with methionine-choline-deficient diets showed that four-week infusion of liraglutide at a dose of 570 mg/kg/day was associated with the anti-inflammatory effect observed as a reduction in the accumulation of M1 macrophages. These cells are crucial players and initiators of the inflammatory process [109].

### 3.3. GLP-1RAs and Reproductive System

PCOS is one of the main causes of anovulatory infertility [84]. Considering the fact that nearly 50% of females with PCOS are overweight or obese, it is believed that excess adipose tissue is the main factor worsening reproductive outcomes. Therefore, the first-line treatment in infertility regarding PCOS women with BMI ≥ 25 kg/m^2^ are lifestyle modifications [110]. However, some women might be resistant to this treatment, and additional practices are required, for instance, pharmacological treatment. The latest studies have shown that GLP-1RAs have provided a positive outcome on body weight loss and glucose level in obese women with PCOS [110]. Except for the effect on weight loss, GLP-1RAs might affect fertility through several different mechanisms [110].

Obesity is associated with abnormalities of the HPO axis. It has been shown that obese women have a lower LH amplitude and mean serum level of LH, which might result in abnormal ovarian follicular recruitment and development, which causes longer follicular phases indicating ovulatory problems [86,111]. It has been noticed that early onset of puberty, menstrual irregularity, pregnancy complications, and spontaneous abortions are additional obesity-related problems aside from infertility [111]. It is possible that some component of obesity influences oocyte metabolism or metabolism of cells that support the developing oocyte [86]. Adipocytes are responsible for aromatizing androgens to estrogens, which affect gonadotropin production and lead to disturbances in menstrual cyclicity and ovulation [84]. Moreover, low LH level during the luteal phase can result in abnormal endometrial development and subsequent embryo implantation. In obese women with the absence of ovulatory dysfunction, infertility has been documented as well [84].

Hormonal dysfunction of visceral adipose tissue causes changes in the adipokine secretion profile, which leads to further disturbances of the HPO axis function. Currently, it is believed that adipokines have a pleiotropic effect on the HPO axis: At the central and peripheral levels. Leptin, apart from increasing the feeling of satiety and inhibiting the feeling of hunger, stimulates the secretion of GnRH in the hypothalamus. Thanks to these functions, signals are integrated, and reproductive functions are linked with the nutritional status of the organism. Both extremely low and extremely high concentrations of leptin inhibit the release of GnRH. The high concentration of leptin in the circulation of obese patients causes the desensitization of leptin receptors in the hypothalamus and then a secondary reduction in the secretion of GnRH and gonadotropins. Leptin influences the hypothalamus through several neurotransmitters, including kisspeptin, neuropeptide Y, γ-aminobutyric acid (GABA), and β-endorphin. At the peripheral level, leptin may directly influence the folliculogenesis process through several mechanisms. Hyperleptinemia induces the transcription of CART in the ovarian granulosa cells, which then reduces estradiol synthesis by inhibiting aromatase. This disturbs the maturation process of the ovarian follicles, reduces the number of ovulations, and increases the percentage of atresia. Similarly, adiponectin under physiological conditions influences the hypothalamic-pituitary axis by stimulating the secretion of FSH and LH. Its decreased concentration in the circulation caused by obesity reduces the secretion of gonadotropins. In ovarian granular cells, receptors for other adipokines were also identified: Adiponectin, visfatin, resistin, chemerin, omentin, and apelin. It has been suggested that in obese patients, these adipokines act peripherally, causing lipid accumulation, lipotoxicity, endoplasmic reticulum stress, mitochondrial dysfunction, and ultimately, cell apoptosis [111,112,113].

GLP-1RAs decrease energy intake and provide adipose tissue reduction [114]. Therefore, they reduce the adverse effects of obesity on the reproductive system in women. Salamun et al. performed a study to evaluate the impact of low-dose liraglutide in combination with metformin on in vitro fertilization pregnancy rate and spontaneous pregnancies in infertile and obese women with PCOS. Results have shown that double therapy was superior to metformin alone in increasing spontaneous pregnancies and in vitro fertilization pregnancy rates, wherein both treatment strategies resulted in comparable reduction in body weight and adipose tissue. Similarly, a reduction in BMI, visceral adipose tissue, and waist circumference had not differed regarding the type of treatment, however, twice as many patients became pregnant using double treatment in an observation period of 1 year [110]. That suggests that an additional mechanism of GLP-1RAs impact on infertility must be proposed.

Hyperinsulinemia plays a fundamental role in the pathogenesis of PCOS [84]. Brüning et al. provided information that knockdown of brain-specific insulin receptors causes infertility due to low LH levels [115]. Obtained effect on LH level was likely due to hypothalamic dysregulation, while pituitary responsiveness was intact. Although most GnRH neurons express insulin receptors, there is no evidence that insulin treatment activates them, and it is proposed that insulin exerts its function on the pituitary by interacting with GnRH signaling pathway in a gonadotropin cell mode [115]. GLP-1RAs have shown effective improvement in IR [116]. Reduction of Homeostatic Model Assessment—Insulin Resistance (HOMA-IR) score promoted by double treatment with GLP-1RAs and metformin has been significant, unlike reduction of HOMA-IR score promoted by metformin treatment only. Hence, the consecutive mechanism of GLP-1A restoring reproductive function is by improving insulin sensitivity leading to a reduced serum level of insulin, and consistently, reduction of LH. Moreover, it has been demonstrated that both combined and single-drug treatment results in increased SHBG secretion, which reduces the bioavailability of androgens [110].

Jungheim and colleagues assessed that despite the superior effect of double treatment with GLP-1RAs and metformin on BMI, IR, visceral adipose tissue, and androgen bioavailability compared to metformin-only treatment, the comparability of these results implies some additional effects of liraglutide on fertility [110]. It has been discovered that GLP-1RAs mRNA is expressed in the cerebral cortex, hippocampus, thalamus, and hypothalamus. Via a specific GLP-1R, GLP-1RAs have the potential ability to regulate GnRH released from the hypothalamic neurons [117,118]. The expression of GLP-1R on the pituitary gland is lower than in the hypothalamus advocating that GLP-1 does not act directly on the pituitary gland to elicit LH release [118].

Treatment with GLP-1 during the proestrus phase in rats doubled the LH serum level and resulted in exerting progesterone in the luteal phase, constantly increasing the number of mature Graafian follicles, resulting in increased fertility [110].

Additionally, it has been demonstrated that short-acting GLP-1RAs improved endometrial function in animal models [110]. It might be mainly due to the effect of GLP-1RAs on oxidative stress and decreasing fibrosis of endometrium. Implantation failure, pregnancy loss, and defective placentation have been reduced as well [110]. Based on the research regarding the impact of GLP-1Ras on the LH ratio presented above, we postulate its modulating abilities. It can either increase LH surge in HPO axis disturbances due to adipose tissue estrogen aromatization or decrease too high LH levels (inducing ovarian androgen secretion) connected with hyperinsulinemia.

### 3.4. GLP-1RAs and Oxidative Stress

Oxidative stress has an important role in the natural history of PCOS [119]. The imbalance between free radicals and their elimination, combined with mitochondrial dysfunction, may participate in the pathogenesis of PCOS [120]. The patients with PCOS have decreased levels of glutathione and total antioxidant capacity [119]. Oxidative stress applies to the PCOS phenotype like obesity, IR, hyperandrogenism, and inflammation [119,121,122]. Furthermore, mitochondrial dysfunction combined with inflammation increases the number of metabolic complications and the risk of CVD [123,124].

Unfortunately, there are no studies on the influence of GLP-1 on oxidative stress in PCOS. The undermentioned studies were carried out on a group of diabetic patients [125,126] or cells [127,128]. In 2015, Rizzo et al. evidenced that the liraglutide treatment decreased oxidative stress and it was independent of the impact of liraglutide on glucose metabolism [125]. GLP-1 increased the ability of beta cells to antioxidize via extracellular regulated kinases pathway and nuclear factor erythroid 2-related factor 2 (Nrf2) translocation [129]. Nrf2 activation also restored impaired insulin secretion in pancreatic beta cells [130]. Moreover, Cai et al. demonstrated that GLP-1 treatment decreased phosphorylation of extracellular signal-regulated kinases ERK1/2 and additionally reversed downregulation of epigenetic factor histone deacetylase 6, a downstream molecular of the EKR1/2 whereby significantly reduced intracellular reactive oxygen species [128]. The above outcomes in reducing oxidative stress may result in protection against endothelial dysfunction and prevent the development of atherosclerosis [128]. GLP-1 therapy can also improve arterial stiffness and reduce LV myocardial deformity [126].

### 3.5. GLP-1RAs Impact on Lipid Metabolism in the Context of IR

Lipid metabolism disorders are closely tied to IR at the molecular level. Increased total cholesterol, triglycerides, and lipoproteins like low and very-low-density lipoprotein have a negative impact on insulin sensitivity.

The treatment of dyslipidemia can be optimized using GLP-1RAs. It is suspected that they modulate miRNA involved in lipid metabolism, such as miR-200b, miR200c, miR-34a, miR-338, and miR-21, and induce activating lipid metabolic enzymes [131]. Based on research from 2018, liraglutide reduces TG, total cholesterol, and non-esterified fatty acids levels. Moreover, lower leptin levels and increased adiponectin levels are observed [132].

It is worth noting that obesity may arise molecularly from the increasing number or size of the adipocytes. However, adults usually have a permanent number of adipose tissue cells, therefore enlargement of adipocytes is the more typical molecular process leading to obesity and it is called adipogenesis [133]. The research conducted on 3T3-L1 cell line adipocytes [134] confirmed the presence of GLP-1R in preadipocytes, as well as in mature adipocytes [133]. Nevertheless, the GLP-1R expression is increased in undifferentiated adipocytes, which are probably more essential when the effect of GLP-1RAs is considered.

The previous studies showed that ligand-dependent transcription factors like PPAR-gamma and CCAAT enhancer-binding protein are involved in the adipogenesis process [135]. GLP-1RAs promote the adipogenesis process, which is proved by increased PPAR-gamma and fatty acid-binding protein expression with lipid cumulation. Interestingly, based on recent studies, adipocytes of obese adults may have decreased capability to differentiate, which can induce cumulation of excess lipids outside of the adipose tissue, like liver or skeletal muscle, and cause IR [133].

When it comes to the impact of GLP-1RAs on lipid metabolism of differentiated adipocytes, controlling expression of fatty acid synthase (FASN) and adipose triglyceride lipase (ATGL) enzymes is a significant issue [133]. FASN is responsible for de novo lipogenesis due to catalyzing the conversion malonyl-CoA into palmitate. The expression of FASN is mediated by PKA and MAPK. GLP-1RAs induce phosphorylation of CREB and ERK and, as a consequence, decrease FASN expression [133]. That may limit hypertrophy of adipocytes and help to reduce the amount of visceral fat. The research from 2018 showed that liraglutide reduces hepatic lipid accumulation, decreases deposition of TG in the liver and mesenteric white adipose tissue because of the downregulation of expression of TG synthesis enzymes involved, like monoacyloglycerol and diglyceride acyltransferases. It is important to note that liraglutide also upregulates these enzymes in the inguinal and cluneal adipose tissue and causes deposition of lower-body subcutaneous fat [132].

ATGL is necessary for lipolysis as it converts triacylglycerols to diacylglycerols, however, the GLP-1RAs have no significant impact on ATGL expression [133]. Recent research from 2021 explains the inducing role of GLP-1 in the expression of fibronectin type III domain containing 5 (FNDC5) gene in pancreatic beta cells. The expression is potentially regulated by the CREB-a transcription factor, which binds the transcriptional start site of the FNDC5 gene. It is suggested that activating the FNDC5 gene in pancreatic beta cells not only stimulates the glucose-dependent insulin secretion but also induces lipolysis and autophagy process [136]. Improved glucose and insulin tolerance results in decreasing serum level of cholesterol, TG, and free fatty acid [136].

To summarize, GLP-1RAs affect the metabolism of adipose tissue in both ways, by directly modulating preadipocyte differentiation and by mediating lipid metabolism processes, resulting in a reduction of lipogenesis and stimulation of lipolysis. This confirms the connection between GLP-1 and dyslipidemia related to obesity. All the above-mentioned mechanisms by which GLP-1 induces insulin sensitivity are presented in Figure 6.

## 4. Discussion and Future Perspectives

Preclinical and clinical trials described in this article clearly indicate that PCOS is a condition with a complex pathomechanism in which metabolic alterations play an important role. IR in this group of patients is primarily the result of obesity and androgens activity, but there are many factors involved in its development [36,38]. Metabolic impairments result in a higher risk of T2DM, NAFLD, and CVD in women with PCOS and significantly worsen their prognosis [137]. Treatment focused on the metabolic aspect of PCOS has been developed for years. Patients are advised to change their lifestyle with increased physical activity and modification of diet involving reduction of saturated fatty acids and simple carbohydrates intake. Pharmacotherapy is mainly based on metformin [1].

In this review, we focused on drugs from the group of GLP-1RAs. They act by glucose-dependent stimulation of insulin secretion in pancreatic islets β-cells and in many other mechanisms [67]. Studies indicate these agents can also reduce IR through direct and indirect effects on the mechanisms involved in its development. Reduction of adipose tissue leads to improvement of its secretion profile, which contributes to increased insulin sensitivity. Inhibition of the immune cell migration and the secretion of pro-inflammatory cytokines reduce inflammation, which is a crucial element in the IR pathomechanism. GLP-1RAs also have a positive effect on lipid metabolism and oxidative stress. They tend to improve fertility either by increasing LH surge in hypothalamus-pituitary inhibition due to estrogen excess connected with obesity or decreasing too high LH levels accompanying hyperinsulinemia.

Patients with T2DM are still the main target group for GLP-1RAs therapy, but approvals for other diseases have appeared. In 2014, liraglutide was the first drug from this group to be approved by the United States Food and Drug Administration (FDA) for weight management in obese patients (BMI ≥ 30 kg/m^2^) without diabetes [138]. On 4 June 2021, the FDA approved semaglutide injections (2.4 mg/week) for weight management in obese patients ≥ 30 kg/m^2^ BMI or overweight patients ≥ 27 kg/m^2^ BMI with at least one weight-related condition, including high blood pressure, T2DM or hypercholesterolemia [139]. Both agents should be used in addition to lifestyle changes involving a reduced-calorie diet and increased physical activity. In SUSTAIN 10 trials, a higher efficacy of subcutaneous semaglutide (1 mg/week) was observed compared to liraglutide (1.2 mg/day) in reduction of body weight and HbA1c. Semaglutide therapy was associated with higher rates of gastrointestinal adverse events (43.9% vs. 38.3%) [140].

Currently (March 2022), GLP-1RAs are not officially approved for use in PCOS. However, as it is described in this review, obesity is a crucial factor in worsening the condition of patients with PCOS. Therefore, body mass reduction provided by GLP-1RAs would be beneficial for them. Moreover, in the last few years, more mechanisms of action of GLP-1RAs have been discovered, including complex effects on IR. In the future, drugs from this group could be approved in PCOS patients even with normal body weight.

GLP-1RAs can be used in combination with metformin. The mechanisms of action of both drugs result in high effectiveness in reducing obesity and IR. Combined therapy enables to lower both agents’ doses, which contributes to decreased incidence of side effects [94,116].

GLP-1 receptors can also be activated by dual GLP-1/glucagon receptor agonists. Glucagon increases satiety, energy expenditure, and blood glucose levels. Dual agonists have the positive properties of glucagon and, due to GLP-1R activation, can prevent glucagon-induced hyperglycemia. Oxyntomodulin is a peptide hormone released from the endocrine L-cells of the gut that activates both GLP1R and glucagon receptors. Oxyntomodulin and other dual GLP-1/glucagon receptors agonists reduce food intake and improve blood glucose levels, which makes them promising treatments for obesity, including in the course of PCOS [141]. Moreover, there are substances which can act as triple GLP-1/glucagon/gastric inhibitory peptide receptor agonists (HM15211) [142]. Both dual and triple agonists are at the early stage of research, but in the future, they may play an important role in the treatment of metabolic disorders, including obesity and PCOS.

Almost all GLP-1RAs are used in subcutaneous injections, which could be a significant disadvantage for many patients. This problem should disappear over time as the number of oral formulations will increase. On 20 September 2019, FDA approved semaglutide as the first oral GLP-1RAs for T2DM therapy. Oral use of the drug is possible thanks to co-formulation with sodium N-[8-(2-hydroxybenzoyl amino] caprylate (SNAC), which prevents digestion of the drug molecules in the gastrointestinal tract and facilitates absorption into circulation [143]. This form proved to be effective in HbA1C and body weight reduction [144].

In conclusion, due to the wide profile of action, GLP-1RAs can be useful in decreasing IR, metabolic disorders, and the resulting CVD risk in women with PCOS. Moreover, improvements in metabolic profile influence hyperandrogenism and ovulatory disorders. Therefore, GLP-1RAs have a positive impact on the overall condition of patients with PCOS. Further studies and clinical trials are needed to better understand the role of GLP-RAs in the disease pathomechanism, optimize doses, and identify the group of patients that will benefit most from that kind of therapy.

## Figures and Tables

**Figure 1 ijms-23-04334-f001:**
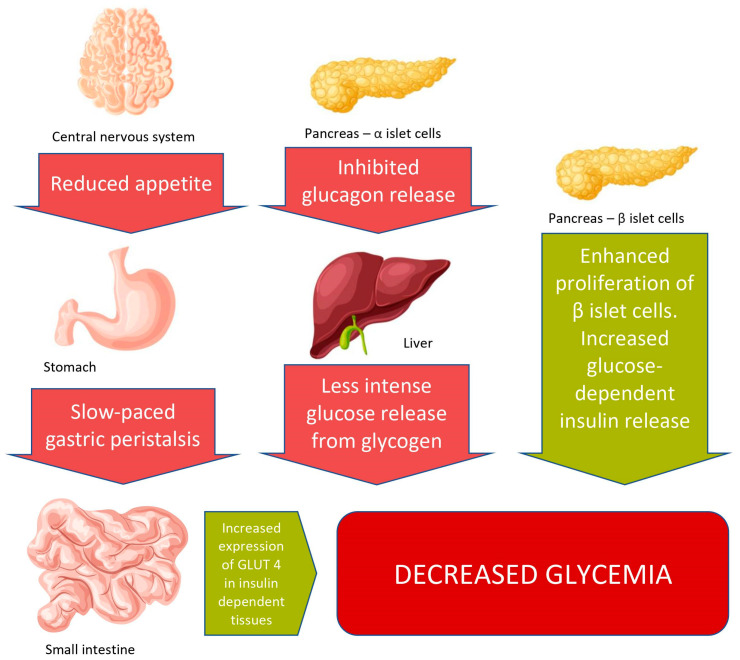
Effects of GLP-1RAs on organs and tissues. The drugs increase glucose-dependent insulin release by acting on pancreatic islets β cells and inhibit glucagon release by acting on pancreatic islets α cells. It leads to decreased glucose release from liver glycogen to circulation. They also enhance the proliferation of β cells. By acting on the central nervous system, GLP-1RAs reduce appetite, which along with slower gastric emptying, leads to reduced food intake. All these effects contribute to a reduction in blood glucose levels. Pictures used to create this figure is designed by macrovector Freepik.

**Figure 2 ijms-23-04334-f002:**
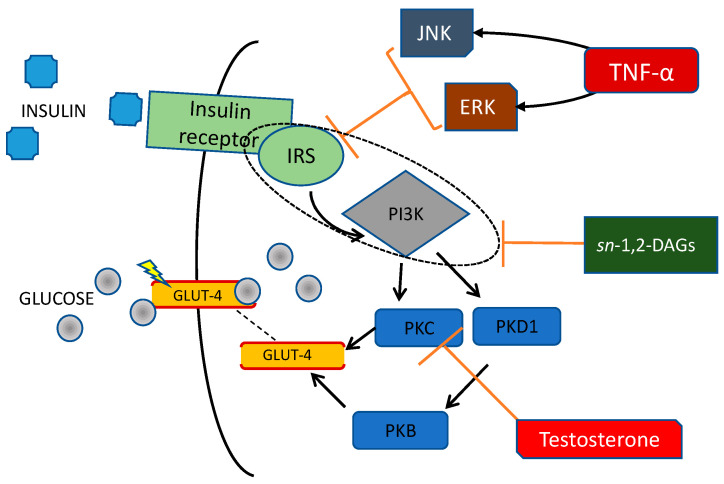
Alterations in the insulin receptor pathway, resulting from the activity of pro-inflammatory cytokines and androgens. Tumour necrosis factor α (TNF-α) affects insulin signaling by phosphorylation of serine in insulin receptor substrate-1 (IRS-1) through activation of several serine kinases, including c-Jun-NH2-terminal kinase (JNK) and extracellular signal-regulated kinase (ERK). It inhibits insulin-induced tyrosine phosphorylation of IRS-1 and downregulates phosphoinositide 3-kinase (PI3K) activity. Decreased adiponectin concentration results in increased membrane sn-1,2-diacylglycerols (sn-1,2-DAGs) activity. It leads to impaired kinases activity and decreased insulin signaling. Testosterone induces insulin resistance in cells by affecting insulin-stimulated phosphorylation of protein kinase C (PKC). All these mechanisms contribute to decreased glucose transporter type 4 (GLUT-4) expression and decreased glucose transport into cells.

**Figure 3 ijms-23-04334-f003:**
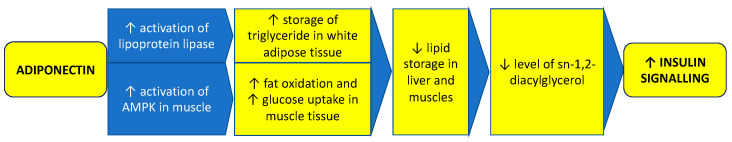
Role of adiponectin in maintaining proper insulin signaling. Adiponectin reduces lipid storage in the liver and muscles, which results in decreased membrane sn-1,2-diacylglycerol levels and increases insulin signaling. Adiponectin mediates these effects by promoting the storage of triglyceride in white adipose tissue through stimulation of lipoprotein lipase and by increasing muscle fat oxidation and glucose uptake by stimulation of 5′ adenosine monophosphate-activated protein kinase (AMPK) in muscle.

**Figure 4 ijms-23-04334-f004:**
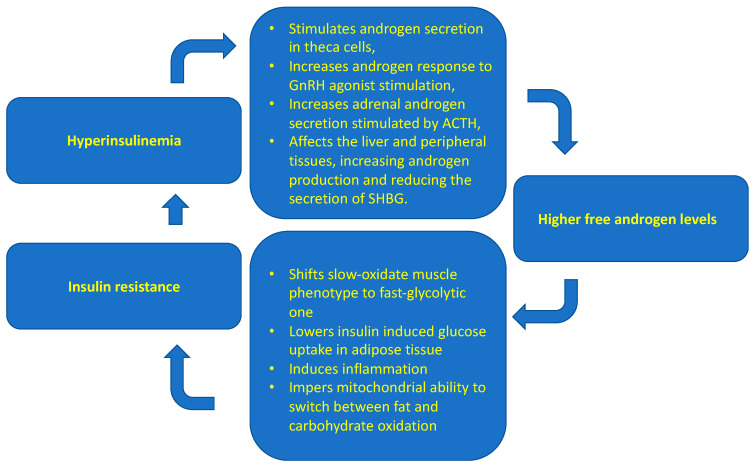
Possible relationships between polycystic ovary syndrome and insulin resistance. The main conclusion is that insulin resistance, the resulting hyperinsulinemia, and increased androgen levels influence each other, and it is difficult to determine which phenomenon occurs first.

**Figure 5 ijms-23-04334-f005:**
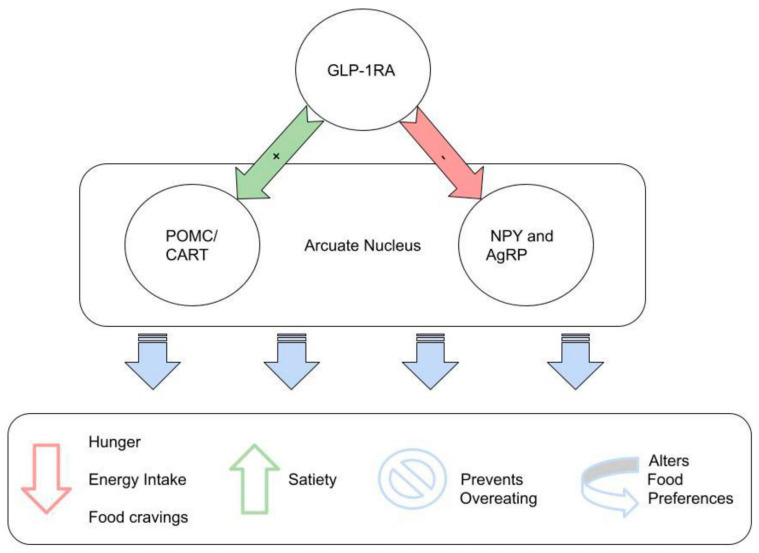
GLP-1RAs reduce feelings of hunger, energy intake, and food cravings. The drugs also increase satiety, prevent overeating, and alter food preference. GLP-1RAs actions are a result of GLP-1R activation on POMC and cocaine-and amphetamine-regulated transcript (POMC/CART) neurons along with indirect inhibition of orexigenic neurons express neuropeptide Y (NPY) and agouti-related peptide (AgRP). All these neurons are located in the arcuate nucleus, which is located in hypothalamus and plays a crucial role in the regulation of hunger and satiety.

**Figure 6 ijms-23-04334-f006:**
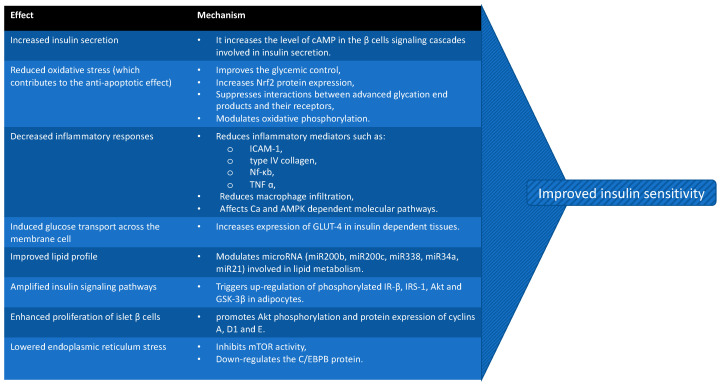
Possible mechanisms by which GLP-1RAs induce insulin sensitivity (cAMP = 3′,5′-cyclic adenosine monophosphate; Nrf2 = nuclear factor erythroid 2-related factor 2; ICAM-1 = Intercellular Adhesion Molecule 1; Nf-κb = nuclear factor kappa b; TNF α = tumor necrosis factor alpha; AMPK = 5′ AMP-activated protein kinase; GLUT-4 = Glucose transporter type 4; mTOR = mammalian target of rapamycin; C/EBPB = a transcription factor).

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
