# Peer review of "The Role of Glp-1 Receptor Agonists in Insulin Resistance with Concomitant Obesity Treatment in Polycystic Ovary Syndrome"

_ijms, 2022, doi:10.3390/ijms23084334_

Round 1
Reviewer 1 Report
It was great pleasure read Review „The role of GLP-1 receptor agonists in insulin
resistance with concomitant obesity treatment in polycystic ovary syndrome”. This review is very well written, contains a lot of interesting information and clearly presents the topic.I have a few minor comments.
- The sentence „ovulatory dysfunction (which appears as endometrial
hyperplasia).” is not precise „endometrial hyperplasia”, please give citation or remove this steatment.
- There is information about higer amount of visceral adipose tissue in PCOS women. Please give citation for this (doi:10.3390/jcm9030732).
- The steatment: „A low-carbohydrate diet (20 to 40 g/d), as well as low-fat diets (40
g of fat per day) were confirmed to be effective [14].” should be improved. In the article authors found no indication that a LGL diet was better than a LF diet for attenuating hyperandrogenism.
- The steatment shoul be improved: ”Minimum 250 minutes of moderate and 150 minutes of vigorous intensityexercise per week are recommended” strictly as in recommendation e.g. „a minimum of 250 min/week of moderate intensity activities or 150 min/week of vigorous
intensity or an equivalent combination of both, and muscle strengthening activities
involving major muscle groups on 2 non-consecutive days/week”
- The authors indicate about metbaolic inflexibility in PCOS, however thera are data showed that PCOS women have normal metabolic flexibility, which could suggest a distinct pathophysiological mechanism for insulin resistance in this group. It could be also added (doi: 10.1507/endocrj.ej13-0115)
- The abbreviation homa- ir should be expanded
Reviewer 2 Report
The publication by Bednarz K. et al. is well written and presents in a concise and understandable way the latest reports on the role of GLP-1 receptor agonists in insulin resistance as an effective treatment of obese PCOS patients.
However I have minor comments:
- Please pay attention to the font size for references numbers starting from chapter 2.
- In chapter 3 there is an incorrect numbering of subsections - please correct it.
- I suggest adding a graphical diagram to Chapter 2 showing the pathomechanism of IR in PCOS, including the role of the most important hormones, cytokines and other particles as well as gene activation and proper signaling pathways. It would also be a chance to replace the borrowed Figure 2 with your own.
- The list of references should be checked once again and the items presented should be standardized.In some of them page numbers (item 17, 41) or the name of the journal (item 28) are missing.
